# Atrial Fibrillation’s Influence on Short Sleep Duration Increases the Risk of Fatness in Management Executives

**DOI:** 10.3390/ijerph19095438

**Published:** 2022-04-29

**Authors:** Richard S. Wang, Shi-Hao Huang, Chien-An Sun, I-Long Lin, Bing-Long Wang, Yao-Ching Huang, Wu-Chien Chien

**Affiliations:** 1Program of Data Analytics and Business Computing, Stern School of Business, New York University, New York, NY 10003, USA; r851126@gmail.com; 2School of Public Health, National Defense Medical Center, Taipei 11490, Taiwan; 3Department of Chemical Engineering and Biotechnology, National Taipei University of Technology (Taipei Tech), Taipei 10608, Taiwan; hklu2361@gmail.com; 4Department of Medical Research, Tri-Service General Hospital, National Defense Medical Center, Taipei 11490, Taiwan; 5Department of Public Health, College of Medicine, Fu-Jen Catholic University, New Taipei City 242062, Taiwan; 040866@mail.fju.edu.tw; 6Big Data Center, College of Medicine, Fu-Jen Catholic University, New Taipei City 242062, Taiwan; 7Department of Computer Science and Engineering, Tatung University, Taipei 104327, Taiwan; cyberpaul@gm.ttu.edu.tw; 8Taiwanese Injury Prevention and Safety Promotion Association, Taipei 11490, Taiwan; 9Graduate Institute of Life Sciences, National Defense Medical Center, Taipei 11490, Taiwan

**Keywords:** atrial fibrillation, fatness, management executive, sleep duration

## Abstract

This study explored whether atrial fibrillation (AF)’s influence on short sleep duration (SD) increases the subsequent risk of fatness in management executives. This study included 25,953 healthy individuals working as management executives with ages ranging from 35 to 65 years (19,100 men and 6853 women) who participated in a qualifying physical filter program from 2006 to 2017 in Taiwan. Men and women who slept < 4 h had a 4.35-fold and 5.26-fold higher risk of developing AF than those who slept 7–8 h normally. Men and women who slept < 4 h had a 6.44-fold and 9.62-fold higher risk of fatness than those who slept 7–8 h. Men and women with AF had a 4.52-fold and 6.25-fold higher risk of fatness than those without AF. It showed that AF induced by short SD increases the risk of fatness. A short SD can predict an increased risk of fatness among management executives in Taiwan.

## 1. Introduction

A majority of American individuals have sleep disturbances, and a small percentage are extremely tired during the day, which affects their daily lives [1,2,3]. Over time, the average sleep duration has decreased. Although our society has changed dramatically, lack of sleep affects our normal routine [4].

The World Health Organization (WHO) recommends the use of body mass index (BMI) to measure fatness, which is calculated by dividing weight (kg) by the square of height (meters). After evaluation by the WHO, a BMI of 25.0–29.9 kg/m^2^ was defined as overweight, and a BMI ≥ 30.0 kg/m^2^ was defined as fatness [5]. Fatness is caused by excessive calorie intake and too little exercise, resulting in the accumulation of calories, which accumulate in the body in the form of fat [5]. In addition to heredity, it is mainly affected by a fatness-producing environment and lifestyle factors. Being overweight and fat are major risk factors for several chronic diseases, including atrial fibrillation (AF), cardiovascular diseases (CVD), and tumors [5,6].

Sleep is an indispensable part of our normal physiological activities; however, under various life and social pressures, insomnia, dreaminess, early awakening, snoozing, and other problems, sleep health has become a topic of concern [7,8,9]. In a busy urban life, in which people use their commute time to catch up, lack of sleep can impair judgment and cause one to be distracted. Making sleep the priority can improve cognitive performance at work [10]. People work long hours, so it is no surprise that work and sleep are closely related [11]. Everything that affects work must also affect sleep [12]. The primary reason that most people cut back on sleep is work [13].

AF is the most common clinical arrhythmia. According to prevalence surveys, approximately 2% of the population has AF, and its incidence increases with age [14]. The mechanism that causes AF is generally believed to be the extremely rapid abnormal discharges in the atrial tissue, resulting in the inability of the atria to contract normally and irregular heartbeat [15]. AF is caused by the atrium replacing the sinus node that normally sends out electrical signals, resulting in a rapid and disorderly heartbeat during the attack [16]. Previously, AF treatment was supported by the three pillars of stroke prevention, heart rhythm control, and ventricular rate control [17]. Projections indicate that the global prevalence of AF will increase 2.5-fold by 2050 [18,19]. One study suggests that adverse outcomes in patients with AF are more common in men and women with low educational level. Therefore, increased attention should be paid to patients with AF and unmarried men with lower educational level to treat AF in a timely manner and prevent its debilitating complications [20].

Previous studies have discussed the relationship between fatness and obstructive sleep apnea (OSA) as they may be related to the growing epidemic of AF and the possible mechanistic links and implications for AF treatment [21]. Fatness is a risk factor for AF, and OSA is highly prevalent in obesity [22]. OSA is associated with AF, but it is unclear whether AF’s influence on short sleep duration increases the risk of fatness. Studies on AF’s influence on sleep duration and fatness in the Asian management executive population are limited. Therefore, we evaluated AF’s influence on short sleep duration and the risk of fatness in management executives. We used the MJ Health Examination Center Database to investigate whether AF’s influence on short sleep duration increases the risk of fatness.

## 2. Materials and Methods

### 2.1. Data Sources

The MJ Health Data Database is a long-term tracking, large-scale, and comprehensive population health database. Since 1994, the health database has collected data on participants who received health examination services at the MJ Health Management Agency. In addition to health data, some participants agreed to donate blood samples as early as 2002. MJ Health Management has four health examination centers in Taiwan, located in Taipei (Northern Region), Taoyuan (Northwest Region), Taichung (Central Region), and Kaohsiung (Southern Region), providing comprehensive services to people in Taiwan and neighboring regions in Asia. The health database stores the actual data of the participants’ previous physical health examination findings. The data can be traced back to the health examination results in 1994. Therefore, many participants in the health database have multiple health examination records. Information, such as type, is collected through health questionnaires, and the human body and biochemical test data are collected through health examinations. These data are continuously collected with no expiration date.

### 2.2. Study Design

The study was conducted on 25,953 healthy individuals working as management executives with ages ranging from 35 to 65 years (19,100 men and 6853 women) who participated in a qualifying physical filter program from 2006 to 2017 in Taiwan (MJ Health Examination Center). The Institutional Review Board of Tri-Service General Hospital approved the study and waived the requirement for personal written informed consent (TSGHIRB No. B-109-39). The flowchart from MJ Health Examination Center in Taiwan is shown in Figure 1.

The health examination items were medical history, height, weight, waist circumference, vision, color discrimination, hearing, blood pressure, and other physical examinations; chest radiography; urine protein and urine occult blood; hemoglobin and white blood cell count; and blood sugar, liver index ALT, renal index creatinine, cholesterol, triglycerides, high-density lipoprotein cholesterol, and low-density lipoprotein cholesterol levels. Each examiner completed a self-review questionnaire divided into basic personal information and important lifestyle information, such as drinking, smoking, sleep duration, and personal health history.

Automatic weight and height scales (HGM) were used to measure body weight and height, respectively. The WHO recommends the use of body mass index (BMI) to measure fatness, which is calculated by dividing weight (kg) by the square of height (meters). After evaluation by the WHO, a BMI of 25.0–29.9 kg/m^2^ was defined as overweight, and a BMI ≥ 30.0 kg/m^2^ was defined as fatness [5].

All examiners were required to report on how long they slept each night to determine the sleep duration. The question was “How many hours do you sleep on average at night?” Sleep duration was divided into <4 h, 4–6 h, 6–7 h, 7–8 h, and >8 h.

AF detection technology (Biomedical’s patented AFIB) can effectively screen out the most dangerous and fatal arrhythmia—AF—and exclude other arrhythmias caused by physiological and human factors. AF can be automatically detected in the blood pressure measurement process through the patented technology of Biomedical. The user measures in the MAM three-average mode. If AF is found, an icon will be displayed on the screen as a method for early medical diagnosis and medical consultation to help patients understand their health status.

Smokers were classified as subjects who currently smoke cigarettes or past smokers and those who never smoke. Moreover, the subject’s alcohol consumption was determined by their self-report, and they were classified as current drinkers, past drinkers, and those who never drink. Every participant self-reported a question on sleep quality measured by three categories: sleeping well, sleeping fairly well, and sleeping poorly.

Participants were asked to indicate their history of diseases such as hypertension. The data were coded as dichotomous “Yes” and “No”. The selection of the relevant variables is constrained by the number of variables. Based on the purpose of this study and the availability of data, there is one measurable dependent variable in this study, namely, obesity. The independent variables comprise sex, marital status, educational level, smoking, alcohol consumption, sleep quality, and hypertension.

### 2.3. Statistical Analysis

Descriptive statistics comprised percentages, mean and standard deviation, the mean and standard deviation for continuous variables, and the sum (proportion) for categorical variables. Multivariate adjusted odds ratio (OR) and 95% confidence interval (CI) were used to analyze the influence of sleep duration on the development of AF and fatness. After adjusting for age, marital status, educational level, smoking, alcohol consumption, and hypertension, conditional logistic regression analysis was performed to assess the association between sleep duration and development of AF and fatness. SPSS version 22 was used in the data analysis (IBM, Armonk, NY, USA). We considered *p*-values < 0.05 as statistically significant.

## 3. Results

This study included 25,953 participants, of whom 19,100 (73.59%) were male with a mean age of 45 ± 7.35 years and 6853 (26.41%) were female with a mean age of 43 ± 6.24 years (Table 1). The age ranged from 35 to 65 years. Moreover, 18.15% of women were single and educated at the university level (42.2%). Marital status included single, married, divorced, and widowed. Educational levels were divided into high school or less, college, university, and graduate school.

Sleep duration was divided into five categories; <4 h, 4–6 h, 6–7 h, 7–8 h, and >8 h. Overall, the most frequent sleep duration for each sex was 6–7 h. However, <1% of both men and women had a sleep duration of >8 h.

Smoking and alcohol consumption were divided into categories as never smoked and never drank, past smoker and past drinker, and current smoker and current drinker. The majority of women never smoked and never drank (95.3%, 94.2%), but one-third of men were current smokers (30.4%). Moreover, 16.1% of men and 13.1% of women had AF.

The risk (adjusted OR) of AF was 4.35 (95% CI = 1.08–1.46) in men who slept < 4 h compared with those with normal sleep duration of 7–8 h (*p* = 0.003) and 5.26 in women (95% CI =1.10–1.60; *p* = 0.002) (Table 2). Sleep duration that is associated with AF was not evident in the other sleep duration groups.

Table 3 shows the multiple linear regression analysis between AF and fatness. In men in the <4 h group, after adjustment for age, fatness had a significant correlation with AF (*β* = 0.52, *p* = 0.002). In women in the <4 h group, fatness was more strongly associated with AF (*β*= 0.68, *p* < 0.001), while differences between men and women in the 4–6 h, 6–7 h, 7–8 h, and >8 h groups were not significant.

Table 4 shows that, in terms of sleep duration, men who slept < 4 h had a 6.44-fold higher risk of obesity than those who slept 7–8 h, and women who slept < 4 h had a 9.62-fold higher risk of obesity than those who slept 7–8 h. In terms of AF, men with AF had a 4.52-fold higher risk of obesity than those without AF. Females with AF had a 6.25-fold higher risk of fatness than participants without AF. As a sensitivity analysis, we performed a logistic regression analysis with different sleep duration categories (<6 h, 6–7 h, 7–8 h, and >8 h). Very short sleep duration (<6 h) had a higher risk of obesity both in male and female (Appendix A).

## 4. Discussion

The results showed that men and women who slept < 4 h had a greater risk of AF than those who slept 7–8 h normally. Fatness was significantly associated with AF in men and women who slept < 4 h. Regarding AF, men and women with AF had a greater risk of fatness than those without AF. Men and women who slept < 4 h had a greater risk of fatness than those who slept 7–8 h. Therefore, AF affects short sleep duration and the risk of fatness.

Sleep duration is associated with P-wave dispersion, QT dispersion, and P-wave duration, which are also predictors of AF [23,24]. Several studies have shown that short sleep duration has negative effects on the endocrine, immunological, and metabolic systems [25,26,27]. Furthermore, insomnia has been suggested as a risk factor that increases the likelihood of developing CVD by inducing autonomic dysregulation and inflammatory pathways [28,29], which is consistent with our study. When AF occurs, the blood output of the heart will be reduced due to the rapid and irregular heart rate, the blood pressure of the patient may decrease, and the patient may have palpitations, chest tightness, dyspnea, wheezing, dizziness, and other symptoms [30,31]. Dyspnea due to AF leads to fatigue, and the associated exercise intolerance can lead to a sedentary lifestyle with a risk of obesity [31], which is consistent with our study.

The result revealed that men and women who slept < 4 h had a greater risk of AF than those who slept 7–8 h normally. Fatness was significantly related to AF in men and women who slept < 4 h. Participants with AF had a greater risk of obesity than participants without AF. Men and women who slept < 4 h had a greater risk of obesity than those who slept 7–8 h. Therefore, AF’s influence on short sleep duration and fatness warrants consideration.

The primary goal of stress management and health promotion programs is to improve health by empowering people to take control of their own lives [32]. Lifestyle choices related to daily health are an integral part of these interventions and critical in assessing their efficacy [33]. To date, concepts such as self-efficacy, self-control, and empowerment have only been assessed through tools that only partially address daily life choices [32]. The Healthy Lifestyle and Personal Control Questionnaire (HLPCQ) is a good tool for assessing the effectiveness of future health-promoting interventions to improve an individual’s lifestyle and well-being [32,33].

A new tool designed to assess general lifestyle is the General Lifestyle Questionnaire (GLQ), a unique tool that simultaneously measures cognitive, physical, social, and other leisure activities and sleeps, diet, and substance use (alcohol and tobacco) [34]. Therefore, this can be a useful tool in clinical practice and research involving understanding how participation in daily activities affects health, well-being, and/or quality of life [31]. Healthy Lifestyle Questionnaire (Cuestionario de Estilos de Vida Saludables, CEVS-II), measures eating habits, resting habits, tobacco, alcohol consumption, other drug use, and physical activity practices [35]. CEVS-II understands the validity and reliability of the Population Healthy Lifestyle Questionnaire, which is an adequate tool for diagnosing healthy lifestyles in the population [35].

There are several limitations to the study. First, sleep duration was self-reported, which may not be accurate. Second, this was a cross-sectional study, causation cannot be inferred, and caution should be taken when interpreting the results. Third, we cannot rule out the left atrial size, underlying heart disease, and other causes closely related to AF and fatness in a few sleep duration subgroups. Finally, the secondary data used for this study may not represent all confounders that may influence sleep duration.

The strong point of this study was the large cross-sectional pool of management executives from 2006 to 2017 from both men and women in their productive age range.

## 5. Conclusions

This study revealed that AF’s influence on short sleep duration increases the subsequent risk of fatness. Short sleep duration predicts an increased risk of fatness among management executives in Taiwan. Therefore, understanding AF’s influence on short sleep duration and preventing modifiable risk factors will help significantly reduce the risk of fatness. The research results will be important evidence for healthcare management to promote not only fatness prevention but all lifestyles. Lifestyle factors are important regulators of physical and mental health and have a great impact on the quality of life and well-being throughout life. Therefore, a proper and comprehensive assessment of lifestyle is essential to detect unhealthy behaviors and prevent their effects on physical and/or mental health. Healthcare professionals should be equipped with tools, knowledge, skills, and competencies in the newly distinguished field of lifestyle medicine, e.g., HLPCQ, GLQ, and CEVS-II, to overcome an increasingly complex medical situation.

Future studies should divide the sample population into more groups according to the combination of obesity and sleep duration and then analyze their relationship with AF.

## Figures and Tables

**Figure 1 ijerph-19-05438-f001:**
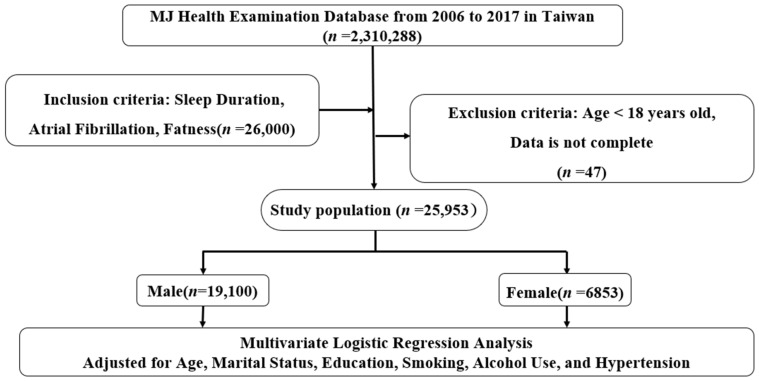
Flowchart of this study.

**Table 1 ijerph-19-05438-t001:** Descriptive statistics of examiners (*n* = 25,953).

Variables	Male (19,100)	Female (6853)	*p*-Value
	*n* (%)	*n* (%)	
**Age** **(mean ± SD, year)**	45 ± 7.35	43 ± 6.24	<0.001
**Marital status**			<0.001
Single	1027 (5.38)	1244 (18.15)	
Married	17,594 (92.12)	5101 (74.43)	
Divorced	415 (2.17)	420 (6.13)	
Widowed	64 (0.34)	88 (1.28)	
**Educational level**			0.785
High school or less	1589 (8.3)	695 (10.1)	
College	4530 (23.7)	1724 (25.2)	
University	7348 (38.5)	2893 (42.2)	
Graduate school	5633 (29.5)	1541 (22.5)	
**Smoking**			0.173
Never	11,176 (58.5)	6529 (95.3)	
Past	2113 (11.1)	76 (1.1)	
Current	5811 (30.4)	248 (3.6)	
**Alcohol consumption**			0.213
Never	14,028 (73.4)	6456 (94.2)	
Past	331 (1.7)	17 (0.2)	
Current	4741 (24.8)	380 (5.5)	
**AF**			<0.001
No	16,012 (83.9)	5950 (86.9)	
Yes	3088 (16.1)	903 (13.1)	
**Sleep duration**			<0.001
<4 h	58 (0.3)	19 (0.3)	
4–6 h	3530 (18.5)	1273 (18.6)	
6–7 h	13,466 (70.5)	4677 (68.2)	
7–8 h	1938 (10.1)	826 (12.1)	
>8 h	108 (0.6)	58 (0.8)	

*n* = sample size. *n*% = sample size percentage. SD = standard deviation, *p* < 0.001.

**Table 2 ijerph-19-05438-t002:** Logistic regression analysis different sleep duration and AF.

Group	Male			Female		
Variable	Adjusted OR	95% CI	*p*-Value	Adjusted OR	95% CI	*p*-Value
7–8 h	1.0	(Reference)		1.0	(Reference)	
<4 h	4.35	2.10–4.60	0.002	5.26	4.81–5.95	0.001
4–6 h	1.29	0.93–1.79	0.13	1.17	0.65–2.09	0.121
6–7 h	1.29	0.93–1.79	0.13	1.17	0.65–2.09	0.341
>8 h	1.3	1.18–1.45	0.999	1.36	0.99–1.85	0.062

Adjusted for age, hypertension, diabetes, dyslipidemia, and BMI estimates. Adjusted OR = adjusted odds ratio, CI = confidence interval, *p* < 0.001.

**Table 3 ijerph-19-05438-t003:** Multiple linear regression analysis between AF and fatness.

Group	Male			Female		
Variable	AF			AF		
	*β*	*R* ^2^	*p*-Value	*β*	*R* ^2^	*p*-Value
<4 h	0.52	0.06	0.002	0.68	0.04	<0.001
4–6 h	0.28	0.06	0.03	0.52	0.09	0.13
6–7 h	0.58	0.05	0.36	−0.28	0.04	0.35
7–8 h	0.34	0.10	0.12	0.61	0.19	0.23
>8 h	0.37	0.30	0.39	0.60	0.14	0.06

AF = atrial fibrillation, *p* < 0.001.

**Table 4 ijerph-19-05438-t004:** Adjusted odds ratios (95% CI) of fatness with sleep duration and AF by sex.

Variables	Male (*n* = 19,100)		*p*-Value	Female (*n* = 6853)		*p*-Value
	Adjusted OR	95% CI		Adjusted OR	95% CI	
**Sleep duration**						
7–8 h	1.000	(Reference)		1.000	(Reference)	
<4 h	6.441	6.441	<0.001	9.622	1.788–51.779	<0.001
4–6 h	1.835	1.377–2.445	0.895	1.941	0.891–4.228	0.769
6–7 h	1.291	0.991–1.681	0.212	1.644	0.809–3.340	0.132
>8 h	1.430	0.561–3.647	0.173	2.845	0.588–13.760	0.324
**AF**						
No	1.000	(Reference)		1.000	(Reference)	
Yes	4.523	3.672–5.299	<0.001	6.258	(4.274–8.277)	<0.001

Age, marital status, educational level, smoking, alcohol consumption, and hypertension were adjusted. Adjusted OR (odds ratio), CI = confidence interval. AF = atrial fibrillation, *p* < 0.001.

## Data Availability

Data are available from the MJ Health Examination Center Database. All or part of the data used in the research were authorized by and received from MJ Health Research Foundation (authorization code: AP_A2020019). Any interpretation or conclusion described in the paper does not represent the views of MJ Health Research Foundation. Due to legal restrictions imposed by the government of Taiwan concerning the “Personal Information Protection Act”, data cannot be made publicly available. Requests for data can be sent as a formal proposal to the MJ Health Examination Center Database (https://www.mjhrf.org/ (accessed on 15 March 2022)).

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
