# Peer review of "Atrial Fibrillation’s Influence on Short Sleep Duration Increases the Risk of Fatness in Management Executives"

_ijerph, 2022, doi:10.3390/ijerph19095438_

Round 1
Reviewer 1 Report
I congratulate the authors on the work done on this relevant subject.
I would like to contribute with some comments, suggestions and questions to the authors:
- Methods section should be improve. Add separate point eg.: Study Design and Setting, Study Population, Screening, Data Collection, Statistical Analysis etc.
- The research results will be important evidence for healthcare management not only to promote fatness prevention but all part of life style. In my opinion is worth to add in discussion part that healthcare professionals should be equipped with tools, knowledge, skills, and competencies in the newly distinguished field of lifestyle medicine e.g. HLPCQ (https://doi.org/10.3390/ijerph18179190) or different (https://www.sciencedirect.com/science/article/abs/pii/S003329841930041X; https://www.elsevier.es/en-revista-international-journal-clinical-health-psychology-355-articulo-validation-spanish-healthy-lifestyle-questionnaire-S1697260021000090).
- Please add practical implication.
Overall, the authors have analyzed the available data to a reasonable conclusion
Reviewer 2 Report
Atrial Fibrillation influenced of little Sleep duration on the risk of Fatness in Management Executives
Richard S. Wang et.al
This study investigated relationships between AF, sleep and obesity.
It appears that the authors have concluded that AF causes sleep duration to be reduced, and subsequently leads to obesity.
The results of this study are interesting, but over obscured by significant language and presentation issues.
Reviewer 3 Report
The authors of this paper intend to explore the relationship between obesity, sleep duration and atrial fibrillation in a relatively healthy large sample population, and the final conclusion is “ the strong association between AF influenced of little sleep duration increases a subsequent risk of fatness. Less sleep duration can predict an increased risk of fatness among management executives in Taiwan”. However, due to major flaws in the study design, I think these conclusions seem untenable.
- The authors suggest that sleep less than fourhrswas associated with atrial fibrillation and fatness was also associated with atrial fibrillation in a few sleep duration subgroups. However, the etiology of atrial fibrillation is multifaceted. This study only adjusted for a few factors such as age, marital status, educational background,high blood pressure, diabetes and BMI, and failed to correct for the left atrial size, underlying heart disease, and other causes closely related to atrial fibrillation . Moreover, there is not enough evidence to support that marital status and educational background can affect the incidence of atrial fibrillation, so why adjust these factors?
- In table 4, the author believes that no matter male or female, sleep durationof < four hrs will significantly increase the prevalence of obesity, but in fact, the sample size of < four hrs is very small, especially the number of women is only 19, such asmall sample size was not sufficient for subsequent statistical analysis.
- The author is discussing the interaction between obesity, sleep time and atrial fibrillation. I think it is more reasonable to divide the sample population into more groups according to the combination of obesity and sleep time, and then analyze their relationship with atrial fibrillation. Otherwise, an interaction between obesity and sleep duration cannot be ruled out.
- There is also a lot of room for improvement in the writing of this article. In the introduction section, the author simply listed the basic conditions of sleep disorders, obesity and atrial fibrillation, but did not further explain why the relationship between the three should be studied, nor did he state what progress has been made in the previous literature in this regard. In addition, although my English is not good enough, I still think that there are many grammatical errors and ambiguous expressions in the writing of this article.
Round 2
Reviewer 1 Report
Now manuscript is ready. Thank you.
